# Intestinal Adaptation upon Chemotherapy-Induced Intestinal Injury in Mice Depends on GLP-2 Receptor Activation

**DOI:** 10.3390/biomedicines9010046

**Published:** 2021-01-07

**Authors:** Anna Billeschou, Jenna Elizabeth Hunt, Aruna Ghimire, Jens J. Holst, Hannelouise Kissow

**Affiliations:** 1Department of Biomedical Sciences, Faculty of Health and Medical Sciences, University of Copenhagen, Blegdamsvej 3, DK-2200 Copenhagen, Denmark; anna.billeschou@sund.ku.dk (A.B.); jenna.hunt@sund.ku.dk (J.E.H.); aruna@sund.ku.dk (A.G.); jjholst@sund.ku.dk (J.J.H.); 2NNF Center for Basic Metabolic Research, Faculty of Health and Medical Sciences, University of Copenhagen, Blegdamsvej 3, DK-2200 Copenhagen, Denmark

**Keywords:** GLP-2, intestine, adaptation, injury, mucositis, mice

## Abstract

Intestinal adaptation is an important response and a natural repair mechanism in acute intestinal injury and is critical for recovery. Glucagon-like peptide 2 (GLP-2) has been demonstrated to enhance mucosal repair following intestinal damage. In this study, we aimed to investigate the role of GLP-2 receptor activation on intestinal protection and adaptation upon chemotherapy-induced intestinal injury. The injury was induced with a single injection of 5-fluorouracil in female GLP-2 receptor knockout (GLP-2R(-/-)) mice and their wild type (WT) littermates. The mice were euthanized in the acute or the recovery phase of the injury; the small intestines were analysed for weight changes, morphology, histology, inflammation, apoptosis and proliferation. In the acute phase, only inflammation was slightly increased in the GLP-2R(-/-) mice compared to WT. In the recovery phase, we observed the natural compensatory response with an increase in small intestinal weight, crypt depth and villus height in WT mice, and this was absent in the GLP-2R(-/-) mice. Both genotypes responded with hyperproliferation. From this, we concluded that GLP-2R signalling does not have a major impact on acute intestinal injury but is pivotal for the adaptive response in the small intestine.

## 1. Introduction

The absorptive function of the small intestine depends on an intact and functional intestinal epithelium. Intestinal injury caused by chemotherapeutic agents, also referred to as gastrointestinal (GI) mucositis, causes an inefficient epithelium, with villus atrophy, crypt cell apoptosis and loss of proliferation. This will lead to impairment of the absorptive capacity of the intestine as well as functional impairment of the barrier [1]. GI mucositis is reported to occur in approximately 40% of patients receiving conventional chemotherapy and 80% of patients receiving high-dose chemotherapy [2]. GI mucositis is associated with diarrhoea, dehydration, abdominal pain and increased systemic infection rate due to inflammation and ulceration of the intestinal mucosa [3]. Despite the severity and incidence of GI mucositis, there is currently no method to prevent this condition and no successful treatment for these serious side effects, and patients mainly rely on symptom reducing treatments, including antidiarrheal agents, antiemetics, antibiotics and rehydration. The development of GI mucositis usually entails a phase of inflammation and epithelial degradation of the tissue with associated ulceration and bacterial migration of the tissue [4]. However, the intestine adapts and renews the tissue over time as a result of epithelial proliferation and differentiation and re-establishment of the local microbial flora, restoring fluid, electrolyte and nutritional balances. The adaptative process is characterized by a proliferation of the remnant stem-cells and during the recovery phase of mucositis, there is an epithelial hyperproliferative repair response [5,6,7,8]. This step is critical in recovery, and is considered a natural compensatory mechanism to the acute injury. However, not all patients with GI mucositis accomplish a full adaptive response and show lack of the compensatory hyperproliferation in the recovery phase, resulting in insufficient intestinal healing, inadequate restoration of functional epithelial surface area with a decreased absorption and digestion of luminal nutrients, showing up as severe body weight loss [5,6,7,8]. A successful adaption is a determining factor for the patients’ long-term survival and for prevention of permanent intestinal failure.

Glucagon-like peptide 2 (GLP-2) is a gut hormone secreted by enteroendocrine L-cells in response to meal ingestion [9]. GLP-2 exerts important digestive control, including regulation of nutrient absorption and stimulation of intestinal growth, increasing the mucosal epithelial surface in the small intestine and thereby its absorptive capacity [10,11]. Despite the multiple roles of GLP-2 in the GI tract, its intestinotrophic effect has attracted the most interest among researchers because of its therapeutic potential. The intestinotrophic effect of GLP-2 was first described in mice by Drucker et al. [9], who had noted a marked proliferation of the small intestinal epithelium in nude mice bearing subcutaneous proglucagon-producing tumours. They identified GLP-2 as a potential factor responsible for the increased small bowel epithelial proliferation [9]. The suggested growth-promoting properties of GLP-2 appear to be highly tissue-specific, with proliferation detected only in the intestine [9] which is in agreement with the distribution of the GLP-2 receptor, subsequently identified [12]. Plasma GLP-2 concentrations have been reported to be significantly increased as a consequence of induced intestinal injury in humans with inflammatory bowel disease [13], and in rats [14] and mice [5] two to five days after induction of mucositis. Besides the enhancement of proliferation, exogenous GLP-2 has also shown to inhibit epithelial apoptosis during both short- and long-term administration of GLP-2 in mice [15].

In a model of total parenterally-fed (TPN) piglets, exogenous GLP-2 as well as enteral nutrition both increased crypt depth and villus height [16], but enteral nutrition was associated with both suppression of apoptosis and stimulation of cell proliferation, whereas GLP-2 only suppressed apoptosis. Taken together, endogenous GLP-2 secreted upon enteral nutrition could be the factor responsible for the proliferative response. However, the mechanisms by which GLP-2 may enhance mucosal repair following intestinal damage and whether the endogenous secretion of GLP-2 is physiologically essential for mucosal repair and protection remains unresolved. In the current study, we examined the effect of endogenous GLP-2 and the importance in mucosal protection and recovery. We tested the hypothesis that deficiency of GLP-2 receptor (GLP-2R) activity would increase intestinal susceptibility to damage by chemotherapy and abolish the adaptive response by preventing the compensatory hyperproliferation. This was tested in a mouse model lacking GLP-2R signalling, in which we measured body weight (BW) loss and small intestinal (SI) weight together with studies of the morphology, histology, myeloperoxidase (MPO) activity, proliferation and apoptotic activity in both the duodenum, jejunum and ileum, three and six days after inducing intestinal injury. We knew from previous findings that at day three mice suffered from the most severe BW loss with severe villus atrophy and that BW and villus height were regained at day six [5].

Indeed, we found that lack of GLP-2R activity abolished the adaptive response, but lack of signalling did not seem to increase susceptibility to chemotherapy in the acute phase.

## 2. Results

### 2.1. Body Weight, Small Intestinal Weight and Morphology

Similar losses in body weight were observed in GLP-2R(-/-) and wild type (WT) mice induced with mucositis during the first three days. This was followed by a weight gain without differences between genotypes (Figure 1A). The SI weight expressed as a percentage of BW was slightly lower in the control GLP-2R(-/-) compared to WT mice (4.36 ± 0.06% vs. 4.00 ± 0.07%, *p* < 0.05). In the acute phase, both genotypes showed significantly reduced intestinal weight (WT 3.38 ± 0.09% vs. 4.36 ± 0.06%, *p* < 0.0001 and GLP-2R(-/-) 3.09 ± 0.09% vs. 4.00 ± 0.07%, *p* < 0.0001) with no differences between genotypes. At day six, the WT mice showed significantly increased SI weight compared to the controls (5.48 ± 0.18% vs. 4.36 ± 0.06%, *p* < 0.0001), however, this was not seen in GLP-2R(-/-) mice (4.21 ± 0.15% vs. 4.00 ± 0.07%, ns) (Figure 1B).

When measuring the villus height we found a reduction in the control GLP-2R(-/-) mice compared to WT mice (559 ± 18 μm vs. 472 ± 17 μm, *p* < 0.01) in the duodenum (Figure 2A). All mice had a reduction in villus height in the acute phase, with no difference between WT and GLP-2R(-/-) mice (duodenum: WT 559 ± 18 μm vs. 355 ± 21 μm, *p* < 0.0001 and GLP-2R(-/-) 472 ± 17 μm, vs. 319 ± 25 μm, *p* < 0.0001) (Figure 2A), (jejunum: WT 323 ± 16 μm vs. 225 ± 12 μm, *p* < 0.0001 and GLP-2R(-/-) 280 ± 13 μm vs. 220 ± 14 μm, *p* = 0.06) (Figure 2B), (ileum: WT 223 ± 9 μm vs. 166 ± 8 μm, *p* < 0.0001 and GLP-2R(-/-) 199 ± 7 μm vs. 139 ± 9 μm, *p* < 0.0001) (Figure 2C). In the acute phase, we also found a reduction in crypt depth, with no difference between genotypes, however, the reduction was only significant in the ileum (duodenum: WT 95 ± 4 μm vs. 90 ± 4 μm, ns and GLP-2R(-/-) 89 ± 2 μm, vs. 81 ± 4 μm, ns) (Figure 2A), (jejunum: WT 92 ± 3 μm vs. 82 ± 5 μm, ns and GLP-2R(-/-) 93 ± 3 μm vs. 78 ± 3 μm, ns) (Figure 2B), (ileum: WT 104 ± 4 μm vs. 82 ± 3 μm, *p* < 0.001 and GLP-2R(-/-) 105 ± 3 μm vs. 80 ± 2 μm, *p* < 0.01) (Figure 2C). At day six, when mice were recovering from the injury, the WT mice had significantly increased villus height in the duodenum (600 ± 15 μm vs. 559 ± 18 μm, *p* < 0.0001) (Figure 2A). The WT mice also had increased crypt depth compared to the controls in both the duodenum (134 ± 7 μm vs. 95 ± 4 μm, *p* < 0.0001) (Figure 2A), the jejunum (130 ± 6 μm vs. 92 ± 3 μm, *p* < 0.0001) (Figure 2B), and the ileum (133 ± 6 μm vs. 104 ± 4 μm, *p* < 0.0001) (Figure 2C). However, in the GLP-2R(-/-) mice there were no increases in villus height and crypt depth (Figure 2A–C). In the recovery phase, both the villus height and crypt depth were smaller in the GLP-2R(-/-) mice than in the WT mice in the duodenum (villus height: 438 ± 21 μm vs. 600 ± 15 μm, *p* < 0.001, crypt depth: 110 ± 5 μm vs. 134 ± 9 μm, *p* < 0.05) (Figure 2A), the jejunum (villus height: 239 ± 13 μm vs. 303 ± 10 μm, ns, crypt depth: 110 ± 3 μm vs. 130 ± 6 μm, *p* = 0.06) (Figure 2B), and the ileum (villus height: 171 ± 8 μm vs. 228 ± 8 μm, *p* < 0.01, crypt depth: 125 ± 9 μm vs. 133 ± 6 μm, ns) (Figure 2C).

### 2.2. Inflammatory Markers and Intestinal Permeability

In the acute phase, both WT and GLP-2R(-/-) mice showed significantly increased severity score in all parts of the SI, but no difference was observed between genotypes. At day six, all animals showed almost full histological recovery (Figure 3A–C). In the acute phase, MPO activity was significantly increased in the jejunum and ileum, with significantly higher levels in the jejunum of GLP-2R(-/-) mice compared to WT mice (19.2 ± 0.8 U/mg vs. 11.9 ± 1.6 U/mg, *p* < 0.001) (Figure 3E,F). In the recovery phase of intestinal injury, the MPO activity was normalized in both genotypes (Figure 3D–F). In the acute phase, FITC-dextran fluorescence in plasma tended to be increased compared to normal in the WT mice (1524 ± 223 ng/mL vs. 713 ± 133 ng/mL, *p* = 0.06) and in GLP-2R(-/-) mice (1580 ± 517 ng/mL vs. 682 ± 121 ng/mL, *p* = 0.13) (graph not shown). There were no differences between genotypes.

### 2.3. Proliferative Activity

Proliferative activity was assessed by counting BrdU positive cells per crypt. In the acute phase, the proliferative activity was returned to the normal proliferative activity with no differences between genotypes. In the recovery phase, proliferative activity was significantly increased compared to controls in all parts of the SI (duodenum: WT 20.0 ± 1.2 cell/crypt vs. 11.4 ± 0.4 cell/crypt, *p* < 0.0001 and GLP-2R(-/-) 19.8 ± 1.3 cell/crypt, vs. 11.1 ± 0.4 cell/crypt, *p* < 0.0001) (Figure 4A), (jejunum: WT 17.3 ± 1.3 cell/crypt vs. 10.7 ± 0.3 cell/crypt, emphp < 0.0002 and GLP-2R(-/-) 17.8 ± 1.5 cell/crypt vs. 10.6 ± 0.4 cell/crypt, *p* < 0.0001) (Figure 4B), (ileum: WT 16.3 ± 0.9 cell/crypt vs. 11.8 ± 0.5 cell/crypt, *p* < 0.0001 and GLP-2R(-/-) 15.5 ± 1.3 cell/crypt vs. 11.2 ± 0.4 cell/crypt, *p* < 0.01) (Figure 4C). We found no difference between genotypes.

### 2.4. Apoptosis

Apoptosis was assessed by measuring caspase-3 activity. In both the acute and recovery phase apoptosis was suppressed, with no differences between genotypes. However, when looking only at healthy mice at day zero, apoptosis was significantly increased in the GLP-2R(-/-) mice in the jejunum compared to controls (1424 ± 165 RLU/mg vs. 2088 ± 231 RLU/mg, *p* < 0.05) (Figure 5).

## 3. Discussion

Since the growth effect of exogenous GLP-2 administered to mice was described by Drucker et al. [9] in 1996, most studies focused on the identification of the exact role of GLP-2 in the small intestine [17,18,19] and highlighted GLP-2 as a potent regulator of intestinal growth [15,20]. Thus, once the rapid and extensive degradation of the hormone by the enzyme dipeptidyl-peptidase-4 had been taken into account, it was acknowledged that exogenous GLP-2 has clear trophic effect on the small intestinal wall in both rats and mice showing increases in villus height, crypt depth and intestinal length [9,15,21,22]. Exogenous GLP-2 has also been demonstrated to prevent chemotherapy-induced intestinal injury in rat small intestine, by preventing villus atrophy [14]. We therefore hypothesized that mice lacking GLP-2 receptor signalling would be particularly susceptible to acute intestinal injury induced by chemotherapy. It has been reported in several studies that plasma levels of GLP-2 increase during intestinal injury, both in humans with inflammatory bowel disease [13], and in rodents after chemotherapy [5,14,23]. Endogenous GLP-2 has been demonstrated to be essential for intestinal regrowth in response to refeeding after a period of nutrient deprivation in both mice [17] and rats [24]. After nutrient deprivation, re-feeding restores intestinal structure via elongation of the crypt-villus axis. Using both human and rat GLP-2 (3-33) to antagonize the GLP-2 receptor, it was demonstrated that endogenous GLP-2 was essential for the maintenance of normal gut structure [17]. Further, blockage of endogenous GLP-2 by GLP-2 immuno-neutralization has been shown to reduce the intestinal growth response in diabetic rats with hyperphagia [25].

Our current results show that GLP-2R signalling did not influence intestinal parameters during acute mucositis. The only difference we found between the GLP-2R(-/-) mice and their WT littermates in the acute phase of mucositis was significantly lesser increased myeloperoxidase activity in the WT mice. The reduced MPO activity could be due to anti-inflammatory properties of GLP-2, but our results do not fully support this since there was a similar increase in histological severity score between genotypes, and the MPO activity was similar between genotypes in both duodenum and ileum. In the recovery phase, both the histological severity score and MPO activity had return to baseline, therefore we have no reason to believe that the state of inflammation in the acute phase had impact on the intestinal adaptation in the recovery phase. Our results showed that GLP-2R signalling did not influence any intestinal parameters in the acute phase. Contrary to this, it was very clear from our results that the normal responses in the recovery phase with increases in SI weight, crypt depth and villus height were absent in mice lacking GLP-2R signalling, proving that intestinal adaptation upon chemotherapy-induced intestinal injury in mice depends on GLP-2 receptor activation. To elucidate the main features of the response we investigated proliferative activity by counting BrdU positive cells, and apoptosis by estimating enzymatic caspase-3 activity. We found that the proliferation rate at day three after chemotherapy was comparable to day zero in the WT mice. This was expected since it has been shown that DNA replication is almost totally abolished at twenty-four and forty-eight hours after 5-FU, but that the number returns to normal at day three [5]. This implies that the activation of signalling pathways towards recovery starts very early after injury. It also explains why apoptosis was significantly suppressed three days after 5-FU. It is evident that cell death after chemotherapy is a very abrupt phenomenon that peaks at around twenty-four h [26,27] and precedes the villous atrophy [28]. We also found the expected increase in proliferative activity at day six, matching the elongated crypts at this time point.

The literature contains multiple examples of contradictory results concerning the effect of GLP-2 on crypt cell proliferation. Arda-Pirincci and Bolkent [29] investigated the effect of exogenous GLP-2 on apoptosis and cell proliferation in a mouse model of intestinal injury induced by tumour necrosis factor-alpha/actinomycin D. It was concluded that pre-treatment with GLP-2 prevented tumour necrosis factor-alpha/actinomycin D-induced injury by significantly reducing apoptosis and markedly increasing cell proliferation [29]. Burrin et al. [16] investigated the effect of both endogenous and exogenous GLP-2 on suppression of apoptosis and stimulation of cell proliferation in total parenteral fed piglets. They found that endogenous GLP-2 was associated with both suppression of apoptosis and stimulation of cell proliferation, whereas treatment with GLP-2 only suppressed apoptosis [16]. Cell growth, differentiation and proliferation are highly regulated by the Wnt/beta catenin pathway [30]. Activation of the Wnt signalling cascade inhibits the phosphorylation of beta-catenin which translocases to the nucleus to activate transcription factors including cyclin D1 and c-myc, which control the G1 to S phase transition in the cell cycle. Entry into the S phase causes DNA replication and ultimately mitosis, which are responsible for cell proliferation [31]. Dube et al. showed that GLP-2 enhanced beta-catenin signalling in the mouse intestine by causing beta-catenin nuclear translocation in proliferative crypt cells increasing c-myc mRNA expression [32]. Due to these earlier findings, we would have expected to find a suppression of proliferative activity and/or an increase in apoptosis that could explain the lack of increase in SI weight, villus height and crypt depth in the GLP-2R(-/-) mice in the recovery phase. However, we found that the GLP-2R(-/-) mice had exactly the same increased proliferative activity as the WT mice.

Having similar proliferative activity does not entirely rule out that the proliferative signal is impaired in the GLP-2R(-/-) mice since we only investigated two time points after 5-FU injection. It is possible that the proliferative activity was suppressed only in the WT mice, e.g., day four leading to the elongated crypts at day five. However, our results do suggest that lack of intestinal re-growth in GLP-2R(-/-) mice is not due to impaired proliferation. Supporting this was the observation that the GLP-2R(-/-) mice had a significant smaller intestinal weight and duodenal villus height at day zero, but a normal proliferative activity.

Small intestinal mucositis is associated with apoptosis in the crypts that precedes hypoplastic villous atrophy and loss of enterocyte height [28]. As mentioned above, GLP-2 has been shown to affect apoptosis [16] and the known increase in plasma GLP-2 after injury could be the stimulus to suppress apoptosis, allowing sufficient adaptation. Impaired suppression of apoptosis due to lack of GLP-2 receptor signalling could explain the lack of compensatory crypt length and SI weight in the GLP-2R(-/-) mice. However, in our hands there was no difference in apoptotic activity after injury. Interpretation of this must take into account that we are only quantifying one of many markers of apoptosis, namely caspase-3. Still we did find that caspase-3 activity was increased in the jejunum of the healthy GLP-2R(-/-) mice where we also found a smaller SI weight and a shortening of villi. Whether these are associated needs further investigation. Likewise, we suggest further investigations to be focused on the implication of GLP-2 in the suppression of apoptosis after intestinal injury.

In summary, our results suggest that GLP-2R signalling is of minor importance in the acute phase of injury, but essential for the recovery. Further studies are required for a better understanding of the mechanism of action, but our results suggest that GLP-2 is not involved in the proliferative response.

## 4. Materials and Methods

### 4.1. Animals

Studies were performed with permission from the Danish Animal Experiments Inspectorate (2018-15-0201-01397) and the local ethical committee. Global GLP-2 receptor knockout (GLP-2R(-/-)) mice [33] were generated by Taconic, using CRISPR/Cas9-mediated gene editing, with Parent Designation: C57BL/6NTac-Glp2rem5153Tac. Animals were bred by heterozygote breeding and wild type (WT) littermates were used as controls. Animals were housed in individually ventilated cages in standard twelve h light, twelve h dark cycle with free access to water and standard chow. In all experiments, we used fifteen-twenty week old female mice.

### 4.2. Experimental Protocol

Thirty-one GLP-2R(-/-) and forty-one WT mice were randomly assigned to receive a single intraperitoneal (i.p.) injection of 5-Fluorouracil (5-FU), 400 mg/kg (Hospira Nordic AB, Stockholm, Sweden) or an i.p. injection with saline. Half of the mice were sacrificed on day three after injections and the rest on day six. Bodyweight (BW) was recorded during the experiment. Mice were fasted for four hours on the day of the experiment and 120 min before sacrifice, and all mice received an i.p. injection of bromo-deoxyuridine (BrdU) 50 mg/kg (B5002, Sigma-Aldrich, Darmstadt, Germany) and fluorescein isothiocyanate dextran (FITC-dextran) (400 mg/kg) (Sigma-Aldrich, Darmstadt, Germany, cat.no. 60842-46-8) by oral gavage (only mice killed in the acute phase and controls). The mice were anaesthetized with a subcutaneous injection of ketamine/xylazine 100 mg/kg ketamine (511485, Ketaminol, Merck, NJ, USA) and xylazine 10 mg/kg (148999, Rompunvet, Bayer, Leverkusen, Germany). When lack of reflexes was established, a laparotomy was performed to expose the abdominal cavity, followed by an incision of the chest to introduce pneumothorax. Blood was drawn from the inferior vena cava and centrifuged (3500 rpm, 15 min, 4 ${}^{\circ }$C). The small intestines (SI) were removed and flushed with saline. Excess saline was removed with a paper towel before the weight was recorded [34]. Sections from the duodenum, jejunum and ileum were collected for histology, morphology, immunohistochemistry and measurement of myeloperoxidase (MPO) activity.

### 4.3. Histology and Morphology Measurements

Segments of intestinal tissue from the duodenum, jejunum and ileum were fixed in 10% neutral formalin buffer (BAF-5000-08A, Cell Path Ltd., Powys, UK) for 24 h. Tissues were dehydrated and paraffin-embedded. Histological sections of 4 μm were cut and stained with Haematoxylin (MHS16, Sigma-Aldrich, Darmstadt, Germany) and Eosin Y solution (318906, Sigma-Aldrich, Darmstadt, Germany). Pictures of each section were taken using a light microscope connected to a camera (Zeiss Axio Lab.A1, Brock and Michelsen, Birkeroed, Denmark) and analysed with Zeiss ImagePro 7 software (Media Cybernetics, Inc. Bethesda, MD, USA). For morphological measurements of villus height and crypt depth, we selected 20 well-oriented villi and crypts. A histological assessment of intestinal damage was conducted using a method described by Howarth et al. [35]. All measurements and evaluations were performed with the observer being blinded with respect to the protocol.

### 4.4. Inflammation

Myeloperoxidase (MPO) activity was measured using extracts from tissue, homogenized in 0.5 procent hexadecyltrimethylammonium bromide (Sigma Chemical Co., St. Louis, MO, USA) in 50 mM potassium phosphate buffer, pH 6.0, as described by Bradley et al. [36]. The homogenates were centrifuged at 16,000 G for 30 min at room temperature and the supernatant was transferred to fresh Eppendorf tubes. For the measurement, 10 μL of the undiluted sample was transferred to a 98-well ELISA plate in duplicates. In total, 190 μL of substrate buffer containing 50 mM potassium phosphate buffers, 0.167 mg/mL O-dianisidine dihydrochloride (20325-40-0, Sigma-Aldrich), and 0.0005% hydrogen peroxide (7722-84-1, Sigma-Aldrich) were added to the wells just before reading. The plates were read every minute at 450 nm using a spectrophotometer (3100-006, Perkin Elmer, Boston, MA, USA) for 8 minutes in total. MPO activity was calculated by measuring the peroxidase reaction of hydrogen peroxide and O-Dianisidine, where one micromole hydrogen peroxide changes the absorbance by 1.13 $\times \;10^{-2}$/min.

### 4.5. Intestinal Permeability

Plasma was diluted in an equal volume of phosphate-buffered saline (PBS) and fluorescence (of the FITC-dextran) was analyzed using an excitation wavelength of 485 nm and an emission wavelength of 528 nm in a SpectraMax iD3 multi-mode microplate reader (Molecular devices, San Jose, CA, USA). The fluorescence measurements were compared to a standard curve of known FITC dextran concentrations.

### 4.6. Immunohistochemistry

The UltraVision Quanto Mouse on Mouse kit (TL-060-QHD, Thermofisher Scientific, CA, USA ) was used for immunohistochemistry. Antigen retrieval was performed by heating sections for 20 min in EDTA buffer, pH 9 (Thermofisher Scientific). The sections were incubated for one hour with monoclonal mouse anti-BrdU antibody (BU20a) (MA1-81890, Thermofisher Scientific) diluted 1:500 with Lab Vision Antibody Diluent OP Quanto (TA-125-ADQ, Thermofisher Scientific). To visualize the BrdU positive cells, horseradish peroxidase (TL-060-QHDM, Ultravision Quanto) alongside its 3,3’diaminobenzidine (TA-125-QHDX, Thermofisher Scientific) substrate were added to the sections. The sections were counterstained with Haematoxylin. Using ImageJ software (https://imagej.nih.gov/ij/, Rockville, MD, USA), proliferation was quantified by counting the number of BrdU positive cells per crypt in ten well-orientated crypts per anatomical section [34].

### 4.7. Apoptosis

Apoptosis was quantified using the Caspase-Glo® 3/7 Assay (G8090 Promega). In short, biopsies from duodenum, jejunum and ileum were homogenized in Luciferase Cell Culture Lysis 5X Reagent (E1531 Promega) (25 Hz, 15 min) (TissueLyser, Switzerland). Samples were centrifuged at 16,000 G for 15 min at room temperature. 25 μL of the diluted (1:10.000) sample were transferred to a 98-well Black ELISA plate and 25 μL Caspase-Glo 3/7 Reagent were added. After 90–120 min incubation at room temperature luminescence were read in a SpectraMax iD3 multi-mode microplate reader (Molecular devices). Since luminescence is proportional to caspase3 activity, results are given in RLU/mg tissue.

### 4.8. Statistical Analysis

Statistical evaluation of changes in body weight (BW) over time was carried out using two-way analysis of variance (ANOVA) followed by a Bonferroni multiple comparisons test. Statistical evaluation of changes in SI weight, morphology, severity score, proliferation, MPO activity, caspase3 activity and FITC-dextran plasma concentrations was carried out using one-way ANOVA followed by a Bonferroni multiple comparisons test. GLP-2R(-/-) were compared to WT at day zero, three and six and mice killed day three and six were compared to their corresponding genotypes without mucositis (day zero). All results are shown as means and standard error of the mean (SEM). Statistical calculations were performed using GraphPad Prism 8 software (La Jolla, CA, USA) and probability values < 0.05 were considered significant.

## Figures and Tables

**Figure 1 biomedicines-09-00046-f001:**
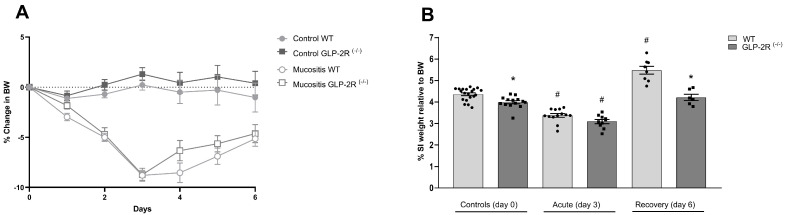
(**A**) Body weight (BW) changes. and (**B**) small intestinal (SI) weight relative to body weight (BW) in GLP-2R(-/-) mice and WT littermates in the acute and the recovery phase of small intestinal injury. *n* = 6–19, * = *p* < 0.05 compared to WT same day, # = *p* < 0.05 compared to same genotype control (day 0).

**Figure 2 biomedicines-09-00046-f002:**
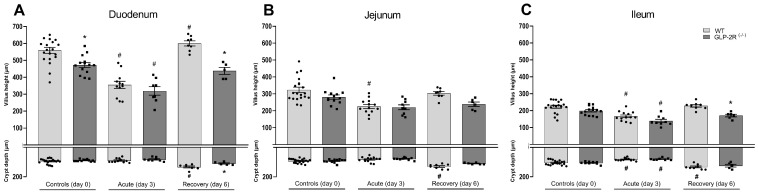
(**A**) Morphology measurement of the duodenum, (**B**) jejunum, (**C**) and ileum in GLP-2R(-/-) mice and WT littermates in the acute and recovery phase of small intestinal injury. *n* = 6–19, * = *p* < 0.05 compared to WT same day, # = *p* < 0.05 compared to same genotype control (day 0).

**Figure 3 biomedicines-09-00046-f003:**
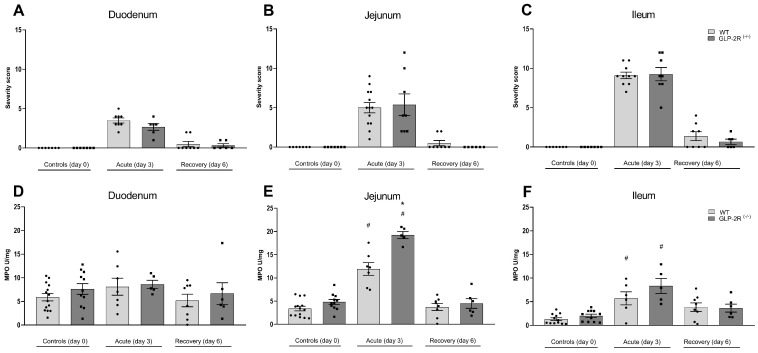
Histological severity score in duodenum (**A**), jejunum (**B**) and ileum (**C**) and myeloperoxidase (MPO) activity in duodenum (**D**) jejunum (**E**) and ileum (**F**) in GLP-2R(-/-) mice and WT littermates in the acute and recovery phase of small intestinal injury. *n* = 6–19, * = *p* < 0.05 compared to WT same day, # = *p* < 0.05 compared to same genotype control (day 0).

**Figure 4 biomedicines-09-00046-f004:**
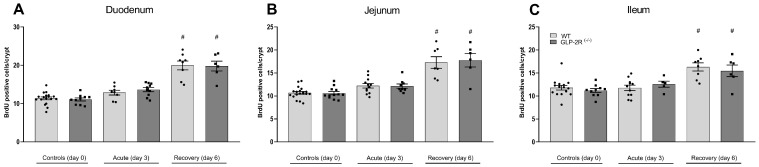
Proliferative activity in duodenum (**A**), jejunum (**B**) and ileum (**C**) in GLP-2R(-/-) mice and WT littermates in the acute and recovery phase of small intestinal injury. *n* = 6–19, * = *p* < 0.05 compared to WT same day, # = *p* < 0.05 compared to same genotype control (day 0).

**Figure 5 biomedicines-09-00046-f005:**
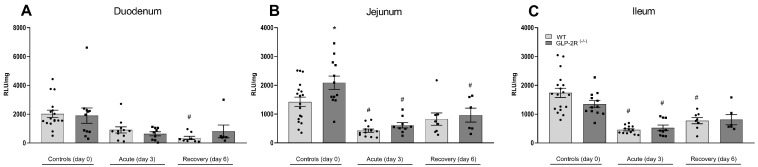
Apoptotic activity in duodenum (**A**), jejunum (**B**) and ileum (**C**) in GLP-2R(-/-) mice and WT littermates in the acute and recovery phase of small intestinal injury. *n* = 6–19, * = *p* < 0.05 compared to WT same day, # = *p* < 0.05 compared to same genotype control (day 0).

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
