# Peer review of "Intestinal Adaptation upon Chemotherapy-Induced Intestinal Injury in Mice Depends on GLP-2 Receptor Activation"

_biomedicines, 2021, doi:10.3390/biomedicines9010046_

Round 1
Reviewer 1 Report
The paper by Billeschou describes small intestinal weight, crypt depth and villus height in Glp-2r deficient mice were not recovered after chemotherapy treatment. The authors have some novel data but some of the methods need to be clarified.
Major Comments:
- The authors just described the phenomenon. There are no data on mechanisms of Glp2r in intestinal adaptation. It is need to clear efficacy of Glp2r signaling.
- The elongation of crypt depth and villus height were seen in only duodenum. So Was the time point (at day 6 after 5-FU injection) sufficient to see these elongation?
- The authors didn’t elucidate what increasing of MPO activity in Glp2r deficient mice effect on intestinal adaptation.
- The authors used one marker for proliferative activity assay. How about Ki67 staining?
Mainor Comments:
- There are double words proliferative activity (under Fig. 3) and apoptosis (under Fig. 4).
- What does the last sentence of paragraph under Fig. 4 indicate? Maybe there are incorrect words or lack of words.
- “Decreased” in 8th sentence of 6th paragraph In discussion section is incorrect.

Reviewer 2 Report
In the manuscript “Intestinal adaptation upon chemotherapy-induced intestinal injury in mice depends on GLP-2 receptor activation”, the authors investigated the role of GLP-2 receptor activation on intestinal protection and adaptation upon chemotherapy-induced intestinal injury; the injury was induced with a single injection of 5-fluorouracil in female GLP-2 receptor knockout (GLP-2R(-/-)) mice and their wild type (WT) littermates. The mice were euthanized in the acute or the recovery phase of the injury; the small intestines were analysed for weight changes, morphology, histology, inflammation, apoptosis and proliferation.
The manuscript is interesting and well written, and the results are clearly presented.
Author Response
Reviewer 2
We thank you very much for the time and effort to review our paper.
Yours Sincerely,
Hannelouise Kissow